# Deep learning with robustness to missing data: A novel approach to the detection of COVID-19

Erdi Çallı[1]*, Keelin Murphy[1], Steef Kurstjens[2], Tijs Samson[2], Robert Herpers[3], Henk Smits[3], Matthieu Rutten[1,4], Bram van Ginneken[1]

1 Diagnostic Image Analysis Group, Radboudumc, Nijmegen, The Netherlands, 2 Laboratory for Clinical Chemistry and Hematology, Jeroen Bosch Hospital, 's-Hertogenbosch, The Netherlands, 3 Laboratory for Clinical Chemistry and Hematology, Bernhoven Hospital, Uden, The Netherlands, 4 Department of Radiology, Jeroen Bosch Hospital, 's-Hertogenbosch, The Netherlands

* erdi.calli@radboudumc.nl

**Citation:** Çallı E, Murphy K, Kurstjens S, Samson T, Herpers R, Smits H, et al. (2021) Deep learning with robustness to missing data: A novel approach to the detection of COVID-19. PLoS ONE 16(7): e0255301. https://doi.org/10.1371/journal.pone.0255301

**Data Availability Statement:** The training and test files are available from Zenodo (doi: 10.5281/zenodo.4461478).

## Abstract

In the context of the current global pandemic and the limitations of the RT-PCR test, we propose a novel deep learning architecture, DFCN (Denoising Fully Connected Network). Since medical facilities around the world differ enormously in what laboratory tests or chest imaging may be available, DFCN is designed to be robust to missing input data. An ablation study extensively evaluates the performance benefits of the DFCN as well as its robustness to missing inputs. Data from 1088 patients with confirmed RT-PCR results are obtained from two independent medical facilities. The data includes results from 27 laboratory tests and a chest x-ray scored by a deep learning model. Training and test datasets are taken from different medical facilities. Data is made publicly available. The performance of DFCN in predicting the RT-PCR result is compared with 3 related architectures as well as a Random Forest baseline. All models are trained with varying levels of masked input data to encourage robustness to missing inputs. Missing data is simulated at test time by masking inputs randomly. DFCN outperforms all other models with statistical significance using random subsets of input data with 2-27 available inputs. When all 28 inputs are available DFCN obtains an AUC of 0.924, higher than any other model. Furthermore, with clinically meaningful subsets of parameters consisting of just 6 and 7 inputs respectively, DFCN achieves higher AUCs than any other model, with values of 0.909 and 0.919.

## Introduction

COVID-19 (the novel coronavirus disease) has disrupted the world since December 2019. Since then, the Reverse Transcription Polymerase Chain Reaction (RT-PCR) test has been the primary testing method to confirm positive cases.

Despite its general acceptance, the RT-PCR test has limited sensitivity, as well as being relatively expensive, and difficult to implement [1]. The RT-PCR requires specialized and expensive equipment as well as trained personnel. Issues such as swab contamination make it

**Funding:** Funding for this study was partially provided by the Botnar Research Centre for Child Health.

**Competing interests:** The authors have declared that no competing interests exist.

difficult to apply in non-optimal circumstances. Even with modern laboratory facilities, it takes at least 2–3 hours to obtain the result of this test. Such constraints are difficult to overcome in the developing world, during localized infection peaks or for mass testing.

As an alternative to the RT-PCR test, meta-analyses [2–4] have shown that routine laboratory test results can distinguish the RT-PCR test outcome, as well as the disease severity and patient mortality risk. Further, some studies [5–10] propose using a chest x-ray to predict the RT-PCR test outcome. In [11] a protocol is defined that combines the chest x-ray interpretation with important laboratory parameters to predict the RT-PCR test outcome. In contrast to the RT-PCR test, the chest X-ray (CXR) and routine laboratory tests are readily available and cheaper to obtain in most scenarios. An overview of various methods to predict COVID-19 diagnosis is provided by [12].

Various machine learning methods have been proposed to make predictions of clinical information from laboratory parameters. For example [13], trains an ensemble of deep neural networks using 41 blood chemistry and cell count results to predict chronological age and sex. They report a mean absolute error of 5.55 years in a test dataset of 6252 patients. In [14], the use of 13 laboratory parameters to predict patient cardiovascular risk is evaluated. The study reports the use of various machine learning algorithms, such as tree-based, regression-based, neural network, or nearest neighbour. Researchers in [15] use various machine learning models to determine the baseline hemoglobin and creatinine levels of ICU patients. They use vital signs (such as temperature and heart rate) as well as 28 routine laboratory parameters. This approach was shown to improve acute kidney injury classification. In [16], 33 laboratory measurements are used as well as age and gender to predict inpatient mortality using various models (such as random forests, boosted trees, or neural networks). This method is compared to the traditional approach of using logistic regression models.

Denoising autoencoders [17] and their variants are used commonly on clinical data. In [18], denoising autoencoders are used on a sleep breathing dataset to improve the cardiovascular disease prediction of a classifier with missing input values. In another study [19], Defines autoencoders to detect coronary heart disease risk prediction using health and nutritional data [20], uses compares stacked denoising autoencoders to FCN, LSTM and SVM models to detect 97 health conditions from 81 clinical features and patient metadata. The same authors later combine two denoising autoencoders, one for high and one for low levels of glycated haemoglobin, to improve the early detection of Type-2 Diabetes Mellitus from 78 features [21].

In this study, we propose a novel architecture, the denoising fully connected network (DFCN), to predict the RT-PCR test result with high accuracy while remaining robust to missing input data. Missing data is an issue in the medical domain, as in many others [22]. There is no global protocol to determine which laboratory or imaging data is collected from a patient. The availability of medical instruments differs per medical facility or may be limited by cost/availability considerations. In the developing world, for example, the range of blood tests or imaging techniques available is typically extremely limited. In any setting data may also be lost or inaccessible due to instrument malfunction or other reasons. Such issues are very common and they pose a problem for machine learning models trained with a specific set of inputs. To overcome this issue, DFCN is designed to function well in the context of missing data, to ensure the utility of the model in most settings globally, including those where laboratory testing and imaging facilities are limited. Unlike the denoising autoencoders used to address missing data in previous works, the DFCN architecture does not include a bottleneck (efficiency constraint) which we demonstrate improves the performance of the model.

During training, we make use of 27 routine laboratory tests combined with the chest x-ray interpretation of a deep learning model. An ablation study is performed to compare the model with related architectures as well as a baseline Random Forest classifier. Performance in the

context of missing data is demonstrated using random subsets of inputs and with two specific subsets, (A and B), selected based on clinical utility and cost efficiency.

DFCN performs significantly better ($p \leq 0.05$) than all other methods in experiments on extensive random subsets of data with 2–27 inputs. In experiments with 1 or 28 available inputs only one other model performs similarly. It obtains 0.924 AUC on the test dataset with all available inputs, which falls to 0.909 or 0.919 respectively if only subset A or B of the parameters is available. We demonstrate that there is no advantage to training these models with the subset of inputs expected at test time. In experiments using test data with subsets A and B of inputs, DFCN trained on all inputs achieves a higher AUC than models retrained with only subset A or B of inputs respectively.

To our knowledge, this is the first time the DFCN architecture has been proposed and evaluated. The useful properties of the architecture are explained through an ablation study and its novel benefits in terms of classification performance and robustness to missing data are demonstrated in this work. It is additionally applied to a globally important problem—diagnosis of COVID-19—where issues of missing input data may be expected.

The remainder of the manuscript is structured as follows: The Data and Methods section defines the dataset, the details of DFCN, and the rest of the models in the ablation study. The Experiments section describes all experiments that were performed, and the results of these are presented in the subsequent Results section. The findings of these experiments are interpreted and discussed further in the Discussion section, which concludes the manuscript.

## Data and methods

In this section, we present the data and proposed methods to combine laboratory parameters with chest x-ray interpretation for prediction of the RT-PCR result using DFCN and other compared methods.

### Data

This study was approved by the Institutional Review Boards of Jeroen Bosch Hospital ('s-Hertogenbosch, The Netherlands), Bernhoven Hospital (Uden, The Netherlands) and Radboud University Medical Center (Nijmegen, The Netherlands). Informed written consent was waived, and data collection and storage were carried out in accordance with local guidelines.

Data was collected from patients attending the emergency department of either hospital with respiratory complaints between 5 March 2020 and 26 April 2020. Data was shared with our institution on 28 May 2020. From those patients, up to 27 laboratory parameters, a frontal chest x-ray, and an RT-PCR test result for COVID-19 were collected. All laboratory parameters were not always available; in Table 1 we present the number of available laboratory parameters per hospital. The distribution of the laboratory parameter values per institute and RT-PCR test results are provided in S1 Fig in S1 File.

The data includes 640 subjects (382 COVID-19 positive) from Bernhoven Hospital (BHH) and 488 subjects (291 COVID-19 positive) from Jeroen Bosch Hospital (JBH), all with confirmed RT-PCR test results. In all experiments, we use the BHH dataset as the training dataset (since it has a larger number of samples) and the JBH dataset as the test dataset. By testing on data from a different institution we demonstrate that our method is generalizable and not overfitted to characteristics of the data from a single institute. The data is publicly available at https://doi.org/10.5281/zenodo.4461478.

Experiments using random subsets of input parameters are used to demonstrate robustness to missing data in general. We additionally define two specific subsets (A and B) which represent more realistic input combinations in practise. Subset A includes the (6) laboratory/

**Table 1. Data used in this study from Jeroen Bosch Hospital (JBH) and Bernhoven Hospital (BHH), divided by RT-PCR result.** Laboratory and Imaging data parameters collected are indicated in the rows. Letters (A, B) are used to indicate which inputs belong to subsets A and B described in Section Data. Numbers of subjects are provided, values in parentheses indicate the numbers of subjects for whom the value was missing or not collected.

| Institute | BHH | | JBH | |
|---|---|---|---|---|
| RT-PCR | Negative | Positive | Negative | Positive |
| **Laboratory Parameter (Subsets)** | | | | |
| Alkaline Phosphatase | 258 (0) | 382 (0) | 193 (4) | 283 (8) |
| Alanine Aminotransferase | 258 (0) | 382 (0) | 193 (4) | 282 (9) |
| Aspartate Aminotransferase | 258 (0) | 382 (0) | 165 (32) | 239 (52) |
| Bilirubin Direct | 32 (226) | 23 (359) | 194 (3) | 289 (2) |
| Bilirubin Total | 258 (0) | 382 (0) | 194 (3) | 290 (1) |
| Creatine Kinase | 258 (0) | 382 (0) | 185 (12) | 271 (20) |
| C-Reactive Protein (A, B) | 258 (0) | 382 (0) | 197 (0) | 291 (0) |
| Absolute Basophil Count (B) | 253 (5) | 378 (4) | 188 (9) | 286 (5) |
| Absolute Eosinophil Count (B) | 253 (5) | 378 (4) | 188 (9) | 286 (5) |
| Absolute Lymphocyte Count (A, B) | 253 (5) | 378 (4) | 188 (9) | 286 (5) |
| Absolute Monocyte Count (B) | 253 (5) | 378 (4) | 188 (9) | 286 (5) |
| Absolute Neutrophil Count (A, B) | 253 (5) | 378 (4) | 188 (9) | 286 (5) |
| D-Dimer | 51 (207) | 12 (370) | 91 (106) | 213 (78) |
| Ferritin (A) | 258 (0) | 382 (0) | 95 (102) | 219 (72) |
| Gamma-Glutamyl Transferase | 258 (0) | 382 (0) | 195 (2) | 291 (0) |
| Hemoglobin | 258 (0) | 382 (0) | 196 (1) | 289 (2) |
| Hematocrit | 258 (0) | 382 (0) | 196 (1) | 289 (2) |
| Potassium | 255 (3) | 375 (7) | 190 (7) | 284 (7) |
| Creatinine | 258 (0) | 382 (0) | 196 (1) | 291 (0) |
| Lactate Dehydrogenase (A) | 249 (9) | 374 (8) | 166 (31) | 239 (52) |
| Absolute Leukocyte Count | 258 (0) | 382 (0) | 196 (1) | 289 (2) |
| Mean Corpuscular Volume | 258 (0) | 382 (0) | 196 (1) | 289 (2) |
| Magnesium | 184 (74) | 224 (158) | 191 (6) | 278 (13) |
| Sodium | 258 (0) | 382 (0) | 196 (1) | 291 (0) |
| Procalcitonin | 222 (36) | 170 (212) | 118 (79) | 266 (25) |
| Thrombocyte Count | 258 (0) | 382 (0) | 196 (1) | 289 (2) |
| Urea | 258 (0) | 382 (0) | 196 (1) | 290 (1) |
| Chest X-Ray (A, B) | 258 (0) | 382 (0) | 197 (0) | 291 (0) |

imaging parameters indicated as important in [11] and likely to be available in medical facilities in most industrialized countries. Subset B includes (7) laboratory/imaging inputs which are relatively inexpensive to obtain and could be acquired by point-of-care machines in a low-resource setting. These parameters were planned for use in a funded research project in Africa [23], based on pilot test results, at the time of writing. The inputs in each of these subsets are indicated in Table 1.

**Validation dataset.**   From the training (BHH) dataset, we derive a validation dataset of 102 patients, 86 of these patients have PA (upright) chest x-rays (43 RT-PCR positive, 43 RT-PCR negative) and 16 have AP (bedside) chest x-rays (8 RT-PCR positive, 8 RT-PCR negative). These represent approximately 20% of the training dataset population and are chosen at random.

## Extracting RT-PCR likelihood score from CXRs

SARS-CoV-2 may infect the lungs and manifest as viral pneumonia, which is usually visible on a chest x-ray as opacities in the lungs. To incorporate information from the chest x-ray in our

system we first train a classifier to predict RT-PCR results based on the chest x-ray image. This classifier is used to produce a likelihood score representing the chest x-ray information. To create a good starting point for this classifier, we use the RSNA-Pneumonia dataset [24] for pre-training.

The RSNA-Pneumonia dataset consists of 26684 chest x-rays acquired in the pre-COVID-19 era. These chest x-rays were annotated by 17 radiologists using 3 labels (Normal, No Lung Opacity / Not Normal, and Lung Opacity). For this study, we split the labeled data into training, validation, and test datasets consisting of 22184, 1500, and 3000 chest x-rays respectively. The training dataset contains 7351 Normal, 10321 No Lung Opacity / Not Normal, and 4512 Lung Opacity cases. The validation and test datasets have equally distributed classes.

We pre-train our convolutional neural network (Resnet 18) on the RSNA-Pneumonia dataset and retrain the last layer of the resulting model on the BHH dataset, to predict a probability for the RT-PCR test outcome.

## Denoising fully connected network

In this study, we propose a novel architecture which we refer to as the Denoising Fully Connected Network (DFCN). The term *denoising* has been used to refer to the objective of reconstructing inputs in the field of Autoencoders [17]. The DFCN architecture is designed such that the primary objective is a classification task (prediction of the RT-PCR result in this case) while the objective of reconstructing masked inputs is applied as a regularizer during training. The model is trained with randomly masked inputs which, combined with this unique regularization method, instills a robustness to missing input data.

The details of the fundamental elements of DFCN are described in the sub-sections below.

**Randomly masked inputs.** The findings of [25, 26] indicate that randomly masking model inputs improves model generalization to unseen data. In the context of internal features of neural networks [27], has shown that dropout (randomly masking the inputs of layers) helps models develop better, more generalized, internal representations. Based on these findings, we identified random input masking as a key feature in developing a model robust to missing inputs. In our training experiments, scalar inputs to the model are randomly set to zero with a specified probability (input masking probability, IMP). Different IMP values are evaluated as described in Section Model Selection: Validation dataset experiments.

**Reconstruction regularization.** In Denoising Autoencoders [17], the objective of input reconstruction is used to pre-train an Autoencoder and hence develop an optimal encoder of the input data. This trained encoder can then be used as the early layers in subsequent tasks such as classification, for example. In DFCN, however, this process is modified to train the decoder and classifier at the same time. The decoding objective is used as a regularizer for the classification task. In this study, we define the reconstruction regularizer based on the L2 sum of the reconstruction errors of the masked and known inputs. This process is illustrated in Fig 1.

**No efficiency constraint.** Denoising Autoencoders are bound to creating so-called *efficient* encodings of the masked inputs. We refer to this property as the efficiency constraint and it is achieved by the use of a "bottleneck" architecture which limits the number of features that represent the input data. In autoencoders [28], it is argued that this bottleneck is necessary to prevent learning a one-to-one mirroring of inputs. When defining DFCN, we bypass this necessity because the reconstruction loss is only applied to the known and masked input values. Hence, the model can not learn such a one-to-one mirroring.

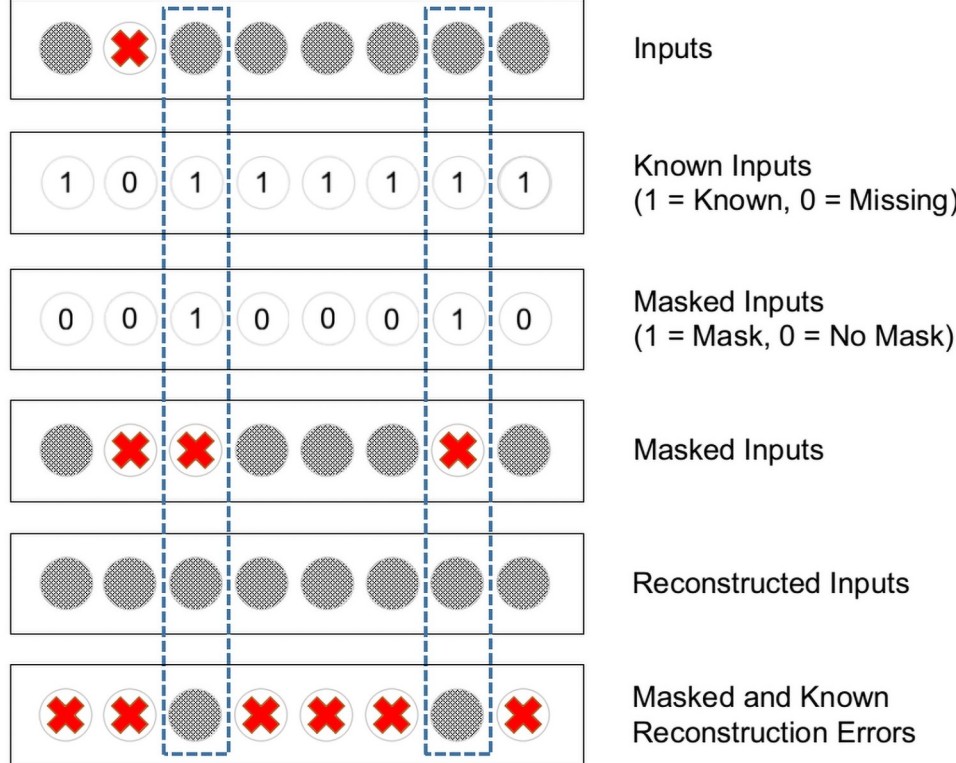

**Fig 1. An illustration of how the reconstruction regularization term is calculated, in combination with input masking.** The top two rows identify data that is missing in our dataset. The subsequent two rows indicate data that is randomly masked at training time. Dashed lines demonstrate that only values which are both known and masked are used in calculating the reconstruction regularization term.

## Ablation study

In this section we describe the ablation study that compares the DFCN to 3 related model architectures to try to identify its important properties. We additionally present results using a Random Forest classifier as a baseline.

Fig 2 illustrates the general high-level architecture that is common among the models which are described in more detail below. The principal idea is to encode the network input and then use this encoding to make a classification as positive or negative. The variants on this architecture used in the ablation study are described in the sub-sections below. All models are trained with various levels of random input masking.

**Denoising fully connected network.** The DFCN uses reconstruction regularization during the training procedure of a fully connected network, as described in detail in section Denoising Fully Connected Network. This model architecture is illustrated in Fig 3.

**Fully connected network.** This model, (FCN), removes the reconstruction regularization that is used during the training of DFCN. It is illustrated alongside DFCN in Fig 3.

**Denoising autoencoder.** As per [17] the Denoising Autoencoder (DAE) uses a two-step training procedure, training firstly the encoder-decoder pair to reconstruct the unmasked inputs, and fixing the weights of the encoder. Next, we use this encoder to extract features from inputs and train a classifier using these features. Compared to DFCN, this model does not make use of the reconstruction objective as a regularizer, but as a pre-training step. Also,

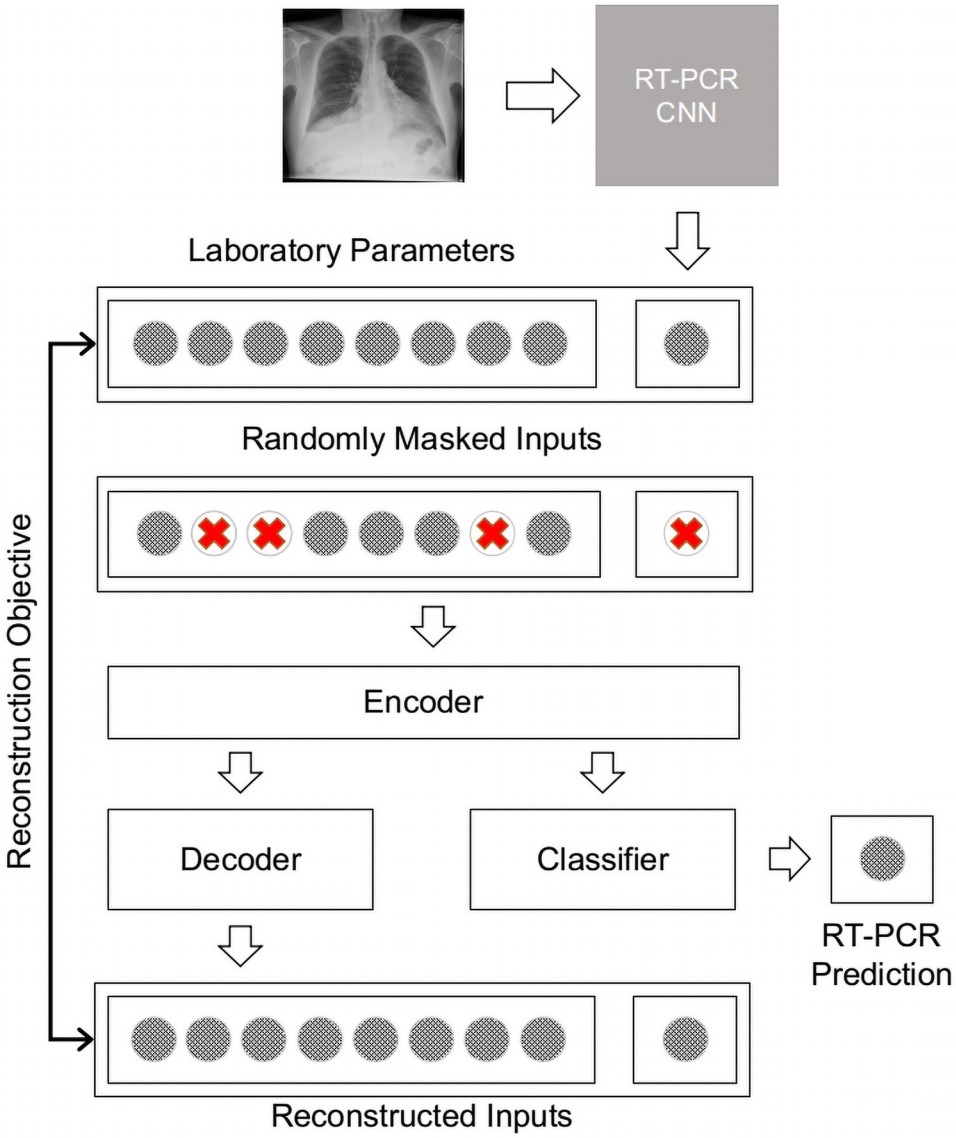

**Fig 2. High level illustration of the architecture underlying the neural networks in the ablation study.** The RT-PCR classifier score from the chest x-ray is extracted using a CNN (top-right). This score is concatenated with the laboratory parameters. During training, inputs are masked by removing a certain percentage of the inputs randomly. The models incorporate an encoder, a classifier (for RT-PCR result prediction), and a decoder to reconstruct the masked inputs.

this model is bound by the efficiency constraint as it is constructed with the typical autoencoder bottleneck architecture. The DAE is illustrated in Fig 4.

**Simplified denoising autoencoder.** This model, (SDAE), uses the same architecture as DAE, but simplifies the DAE training procedure by removing the initial encoder-decoder training step and using the reconstruction objective as a regularizer while training the classifier. The training process is therefore the same as DFCN, but the model is bound by the efficiency constraint imposed by the bottleneck architecture. This model is illustrated in Fig 4.

**Random forest.** Random forest (RF) is a machine-learning classifier introduced by [29] as an ensemble of decision trees. Random forest is known to be robust to noise and is

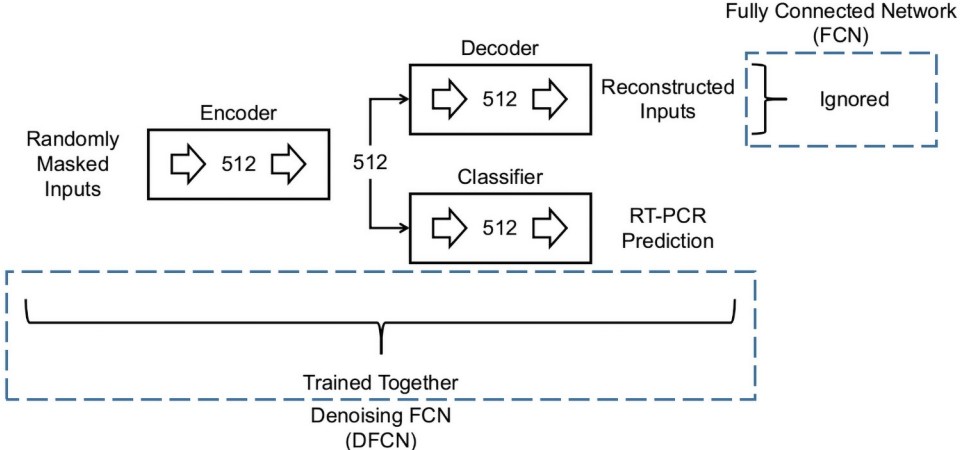

**Fig 3. Architectures of DFCN and FCN.** DFCN is trained using randomly masked inputs and uses the reconstruction of the inputs as a regularizer during the training process. FCN ignores the reconstruction objective. Numbers of neurons are indicated alongside the fully connected layers (large right arrows).

considered a very strong classifier. It is included here as a baseline method which does not incorporate deep learning. The decoder and associated reconstruction objective that is shown in Fig 2 do not apply to this classifier.

**No input masking.** All models used in this ablation study are trained using various probabilities of random input masking. This includes the random masking probability of 0, or in other words, No Input Masking (NIM). Training without input masking alters the behaviour of some of the models, since it effectively disables the reconstruction regularizer. Models trained without any input masking will be referred to with NIM as a prefix to the model name.

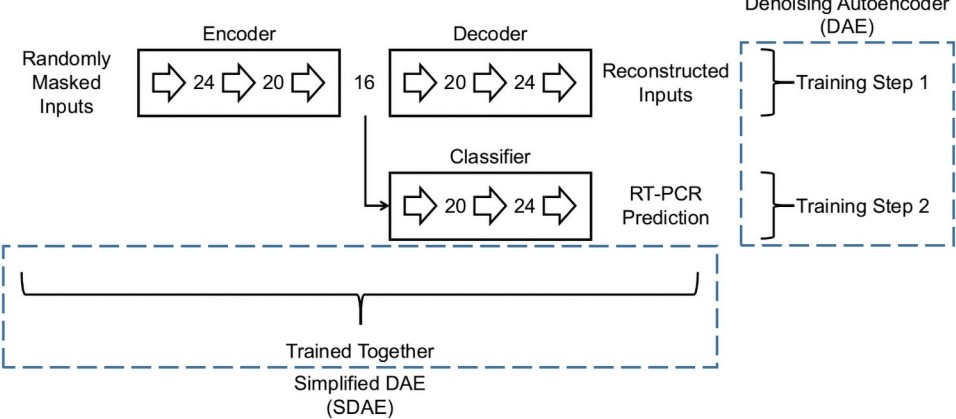

**Fig 4. Architectures of DAE and SDAE.** DAE trains the encoder and decoder first with the objective being to reconstruct masked inputs. The classifier is trained in a subsequent step using the features extracted from the encoder. In SDAE the decoder and classifier are trained together and the reconstruction loss is used as a regularizer for the classification. The numbers of neurons are indicated alongside the fully connected layers (large right arrows). Both models incorporate the efficiency constraint by the use of a bottleneck architecture.

### Evaluation metrics

We use the area under the receiver operating characteristic curve (AUC) to represent the performance of an individual model. The Scikit-learn [30] implementation of the AUC metric is employed. To determine whether an AUC improvement is statistically significant, we use DeLong's test [31]. A threshold of 0.05 ($p \leq 0.05$) is used to indicate statistical significance.

## Experiments

This section describes the experiments designed to evaluate the performance of DFCN and compare it to the other methods in our ablation study. Detailed training settings (such as pre-processing, hyperparameters, optimization algorithms) are provided in S1 Text in S1 File.

### RT-PCR test prediction from CXRs

A Resnet-18 [32] network, pre-trained on the ImageNet dataset [33] is trained first on the RSNA-Pneumonia dataset [24]. To prevent overfitting, we re-initialize and fine-tune only the last layer of this model on the training dataset from BHH to predict the RT-PCR test results and apply softmax activations to obtain likelihood of positive and negative outcomes.

### Model selection: Validation dataset experiments

In this section we describe the process used to select optimal models based on performance on the validation dataset. We use the 27 laboratory parameters as well as the chest x-ray score from the model described in Section RT-PCR test prediction from CXRs to train all models, DFCN, FCN, DAE, SDAE and RF using the training dataset from BHH. The models are trained with input masking probabilities (IMP) of 0.0 (no input masking (NIM)), 0.1, 0.2, 0.3, 0.4, 0.5, 0.6, 0.7, 0.8 and 0.9, which results in a total of 50 trained models.

Next, we compare the performance of all trained models in the context of robustness to missing data. For this purpose a heavily masked version of the validation dataset is constructed. To compose this dataset, we first construct a set of combinations of input parameters masking 0–27 of the 28 input parameters. All 28 combinations with length 1 are included, as well as the single combination with length 28. For each of the combination lengths 2–27, 15 random combinations are chosen. This results in 419 (28 + 1 + (15*26)) unique input parameter combinations. Applying these combinations to the validation dataset (and ignoring the samples that end up having no values), we obtain a heavily masked dataset of 13,596 samples (5,668 positive). We evaluate the AUC of each model using this heavily masked version of the validation dataset to determine performance and robustness to missing data.

For each of the five methods (DFCN, FCN, DAE, SDAE, RF) the model with the highest AUC is selected and the model trained with that IMP value is used in further experiments. The NIM (no input masking) version of each method is also selected, resulting in a total of 10 models for further experimentation.

### Ablation study on independent test dataset

The 10 models selected based on performance on the heavily masked version of the validation dataset are next applied to the test dataset from a different institute. To evaluate their robustness to missing data we construct a heavily masked version of the test dataset, in a similar way as described for the validation dataset. For a more thorough evaluation, however, when constructing random input combinations of length 1–28, we now select 1,000 random combinations of inputs (or the maximum possible number where 1,000 is not feasible) for each specified combination length. This results in 23,813 different combinations of inputs.

For each model, we calculate the AUC on all test dataset samples with *n* available inputs for *n* from 1 to 28. These values are used to indicate performance robustness as the number of available inputs changes. The mean AUC over all 28 values is also calculated as an overall performance indicator. We use DeLong's test to evaluate the significance of the AUC differences between models for each number of available inputs.

### Clinically relevant subsets A and B

In this section we experiment with reduction of the input parameters to specific subsets (A and B) which are chosen for clinical/cost reasons. The inputs used for these subsets are indicated in Table 1. For these experiments, we limit the test dataset to only those patients with all of the inputs available. The models trained without input masking (NIM) are excluded based on their observed poor performance, particularly with reduced numbers of inputs. The AUCs of the 5 remaining models are calculated on the test dataset limited to the inputs in A and B respectively.

Furthermore, since a typical machine-learning approach involves training with a fixed set of data inputs which are expected to be available at test-time, we retrain the 5 optimal models using only subsets A and B of inputs respectively. We train these models using various input masking probabilities and select the best performing models using the performance on the validation dataset. The AUCs obtained by these retrained models, on the test datasets with input parameters from A and B, is compared to the performance of our models on the same data. This determines whether training and testing with identical sets of inputs can improve on our more generalizable models which are designed to adapt to the available inputs.

## Results

### RT-PCR test prediction from CXRs

The Resnet-18 achieves 0.855 AUC in the RSNA-Pneumonia dataset for detection of pneumonia in chest x-ray. This model trained further to predict the RT-PCR test result from the chest x-rays of the training dataset achieves 0.808 AUC in the test dataset from a different institution. This result is comparable with the performance (0.810 AUC) reported by [5].

### Model selection: Validation dataset experiments

Table 2 shows the results of Experiment Model Selection: Validation dataset experiments in which we select models based on the AUCs achieved on the heavily masked validation dataset (13,596 samples, 5,668 positive). In this table the AUC values are presented for the most optimal IMP settings per architecture and for the models trained with no input masking (NIM). An extended version of this table with results for all IMP values is provided in S2 Table in S1 File.

**Table 2. Results on validation set data.** Five models are trained using various input masking probabilities (**IMP**). Each resulting model is validated using the heavily masked validation dataset of 13596 samples (5668 positive) to evaluate their performance in the context of missing input data. AUC values for the optimal training IMP are shown, along with those achieved with no input masking (**NIM**). Bold font indicates the highest AUC in the table. Results for other IMP values are provided in the S1 File.

| | | DAE | SDAE | FCN | DFCN | RF |
|---|---|---|---|---|---|---|
| **Most Robust Models** | **IMP** | 0.2 | 0.4 | 0.6 | 0.6 | 0.7 |
| | **AUC** | 0.815 | 0.834 | 0.838 | **0.843** | 0.831 |
| **No Input Masking (NIM)** | **IMP** | 0.0 | 0.0 | 0.0 | 0.0 | 0.0 |
| | **AUC** | 0.811 | 0.746 | 0.799 | 0.763 | 0.796 |

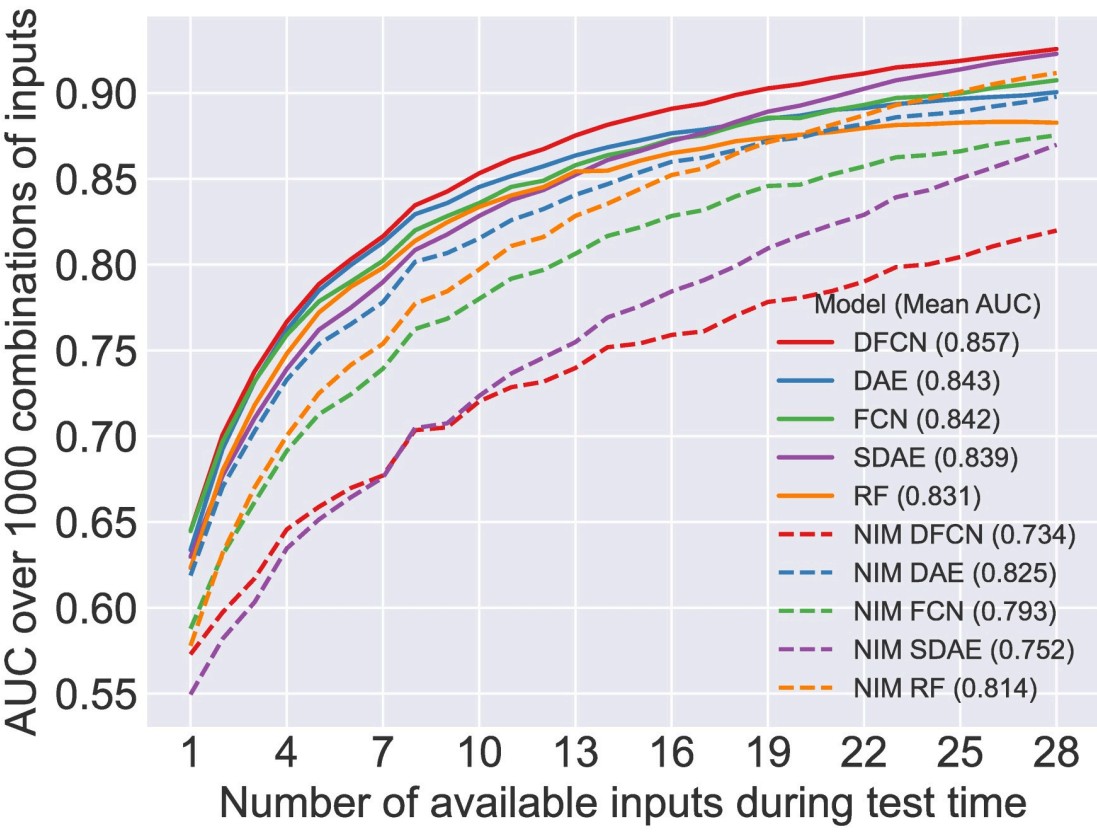

**Fig 5. Comparison of the 10 selected models on test datasets with an increasing number of available inputs.** Each test dataset is composed of 1000 random combinations of *n* inputs (n from 1–28) as described in experiment Ablation study on independent test dataset. Models are trained without input masking (NIM), or with the optimal input masking as per Table 2. Mean AUC across all numbers of available inputs (1–28) is provided in the legend.

All of the models with optimal IMP setting perform better than their counterparts trained with no input masking. The DFCN trained with 0.6 IMP outperforms all other models at their optimal IMP setting, achieving 0.843 AUC on the heavily masked validation dataset. The 10 models identified in Table 2 are selected for further experimentation on the test dataset.

### Ablation study on independent test dataset

The results of applying the 10 models presented earlier in Table 2 to the test dataset from a different institute are presented in this section. The test data is configured into 28 sets with combinations of 1–28 available input parameters (as described in experiment Ablation study on independent test dataset). The AUCs achieved by the selected models on each of these sets are plotted in Fig 5. DFCN clearly outperforms other methods with a higher AUC at all points and a mean AUC of 0.857 across all numbers of inputs, demonstrating robust performance in the context of missing data.

The statistical significance of the differences in these AUC values is calculated and summarised in Fig 6. Here it is illustrated that for 2 to 26 inputs, DFCN performs significantly better than every other model. With other numbers of inputs (1 or 27) its performance is matched (but not improved upon) by other models (FCN (1 input), SDAE (27 inputs)). For 28 inputs, its performance is not significantly different to SDAE, FCN, and NIM RF, however there are relatively few data points (250) with all 28 inputs, making significance difficult to achieve.

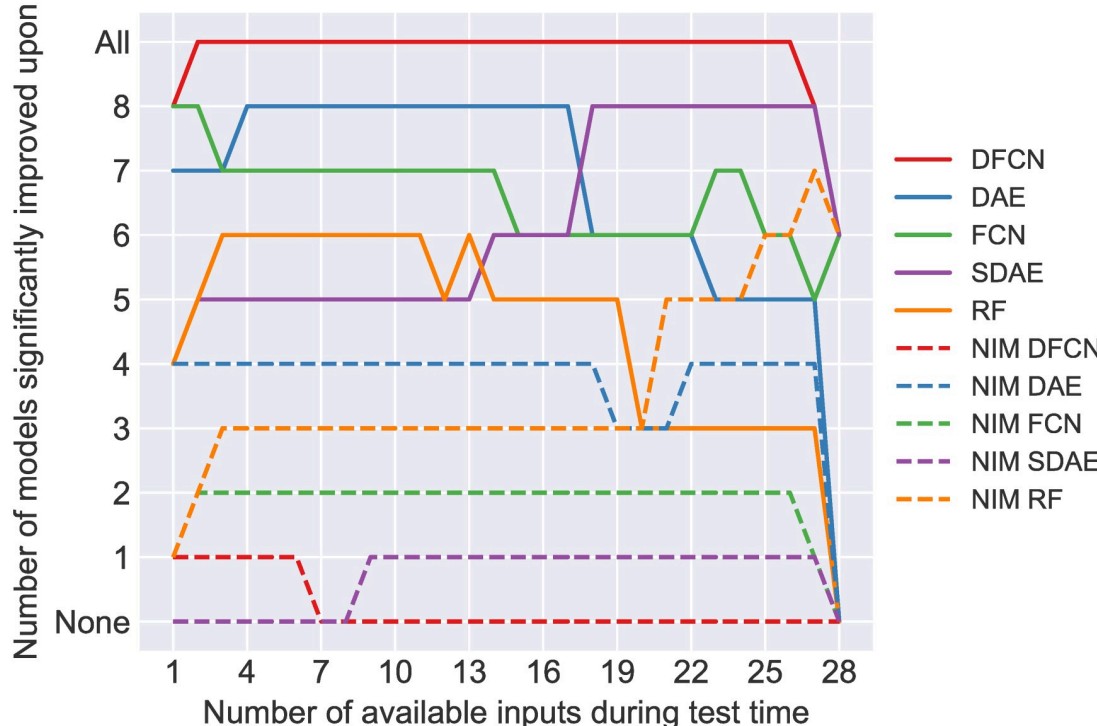

**Fig 6. Statistical significance of differences in the AUC values presented in Fig 5.** Significance is determined by DeLong's test with ($p < 0.05$). The y-axis value indicates how many of the other models were statistically worse in terms of AUC. (Models with the same y value are statistically equal in performance).

Conversely, NIM DFCN is one of the worst models, performing significantly worse than all of the other models from 8 inputs upwards, and only better than NIM SDAE with smaller numbers of inputs. In general, the models trained with no input masking (NIM) perform significantly worse than those trained with input masking, with the exception of NIM RF which outperforms RF, DAE and FCN at various points with >20 available inputs.

### Clinically relevant subsets A and B

Table 3 shows the results of experiments using datasets with inputs reduced to subsets A and subset B. In the top section of the table, as a benchmark, the performance of each model,

**Table 3. Comparison of models on the test dataset from a different institution using specific input subsets (subsets A and B) selected for clinical/cost reasons.** The top section indicates the benchmark performance when both training and test-sets include all inputs. In the central section performance on test-data with only subset A of inputs is shown. Models are trained with all inputs and with only subset A of inputs. The lower section of the table repeats these experiments for subset B. Bold font indicates the highest AUC value in the table section. * indicates AUCs that are significantly lower than the highest in that section ($p < 0.05$).

| Available Inputs | | # Samples | AUC | | | | |
|---|---|---|---|---|---|---|---|
| Training | Test | (# Positives) | DAE | SDAE | FCN | DFCN | RF |
| All | All | 488 (291) | 0.902* | 0.917 | 0.909* | **0.924** | 0.870* |
| All | Subset A | 258 (179) | 0.879* | 0.891* | 0.901 | **0.909** | 0.875* |
| Subset A | | | 0.886 | 0.904 | 0.908 | 0.901 | 0.887* |
| All | Subset B | 474 (286) | 0.897* | 0.893* | 0.906* | **0.919** | 0.883* |
| Subset B | | | 0.897* | 0.912 | 0.910* | 0.912* | 0.894* |

trained with optimal input masking and on all of the available inputs during training and testing is shown.

The central part of the table shows the results for experiments with subset A as the test data input. The highest AUC (0.909) is obtained by the DFCN trained with all inputs. It is shown to be statistically better than four other models in these experiments, and is considered equivalent to the remainder, including the DFCN trained with the input parameters of subset A (AUC = 0.901).

In the last part of the table the results of the same experiments using inputs from subset B are shown. Again, the highest AUC is achieved by DFCN trained with all inputs (0.919). This is statistically better than 7 of the 9 remaining results in this section, including DFCN trained with subset B inputs (AUC = 0.912). It is notable that statistical significance is more difficult to achieve in all experiments shown in Table 3 since the number of data points is small compared to those in the experiments of Fig 5 (with $x$ in the range [2, 27]).

The ROC curves of the highest performing model from each section of Table 3 are provided in Fig 7. The p values for all model comparisons in these experiments are presented in S3 Table in S1 File.

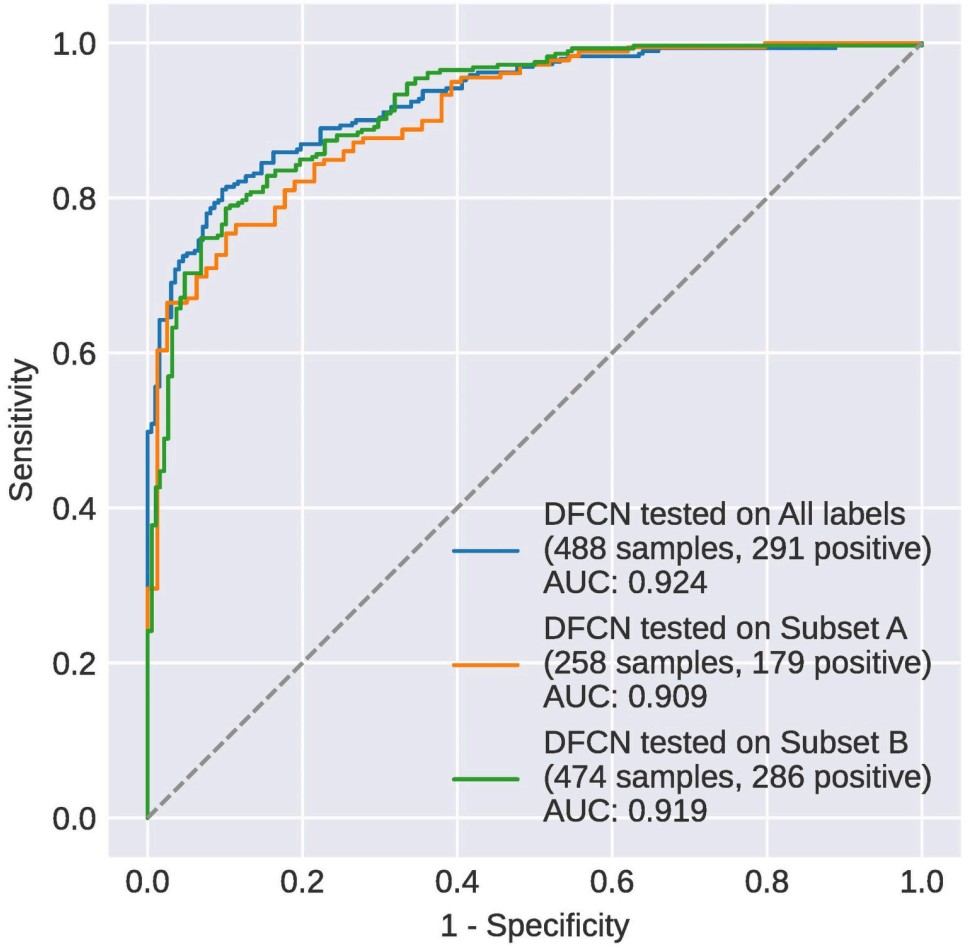

**Fig 7. Comparison of the ROC curves of DFCN for all available inputs, for the subset A and the subset B.**

## Discussion

In this work we have proposed a novel deep learning architecture, DFCN, designed to achieve a robust performance in the context of missing input data, a common issue in the medical domain. The model has been evaluated against other architectures in an ablation study, studying performance and robustness for diagnosis of COVID-19 from laboratory and imaging input parameters. In extensive experiments with randomized subsets of available inputs, DFCN has been demonstrated to achieve a significantly superior performance compared to all other tested models.

The ablation study in this work investigates a number of properties of the DFCN, including the reconstruction regularization term, the use of a "bottleneck" structure associated with autoencoders, and the two-step training process typically associated with autoencoder based classification. It further thoroughly investigates the role of input masking during training of all models and the performance of a baseline Random Forest classifier.

By comparing the SDAE and DAE models, we show that using the reconstruction objective as a regularizer rather than a pre-training step improves the model performance when most of the inputs are available. Both of these models are restricted by the efficiency constraint, to encode data in a limited number of features. In the DAE model these features contain information relevant to reconstruction, while in SDAE the features are related to both reconstruction and classification. We theorize that reconstruction information is more important when few inputs are available, but becomes less important as more inputs are added. This explains why DAE performs better with few inputs, but is surpassed by SDAE when $> 18$ inputs are present.

Further, our comparison between DFCN and FCN demonstrates that the inclusion of the reconstruction regularizer is important and significantly improves the model performance for all numbers of inputs $>1$.

The comparison between DFCN and SDAE shows that the "bottleneck" architecture, or efficiency constraint, does not confer any advantage in this task. Its removal improves the model performance significantly for experiments at all levels of missing inputs, and particularly where the numbers of available inputs are smaller. Removing the efficiency constraint likely provides more capacity for the DFCN to learn specific representations for different combinations of inputs.

Input masking has been demonstrated to contribute substantially to the model performance and robustness to missing data, through the comparison of each model with its NIM (no input masking) version. The optimum level of input masking probabilities varied between models from 0.2 to 0.7 suggesting that this value is dependent on characteristics of the model design and architecture. The Random Forest classifier was the only model where the NIM version outperformed the version with input masking, achieving a better performance when $>20$ input parameters were available. In contrast to the performance of DFCN, the results achieved by NIM DFCN were the worst in most experiments. This demonstrates the importance of applying input masking when the reconstruction objective is in place.

Two subsets of the 28 available inputs (A and B, with 6 and 7 inputs respectively) were selected based on their clinical utility and their practicality in terms of cost and ease of use. We have demonstrated that in a setting where only subset A or B of input data is available at test time, the DFCN, trained on all data, achieves the highest AUC among all models, including the DFCN trained with only the specified subset of inputs. This is the case for both subsets A and B, tested in separate experiments. Although statistical significance was not achieved in all comparisons, we believe that this may be related to the relatively small numbers of samples involved, compared to the experiments illustrated in Fig 5. Even assuming that there is no statistical difference between training on all inputs and training on the specified subset, these

experiments demonstrate that there is no advantage to the latter training scheme, which, in practise, would require training new models for each additional setting where the model would be deployed with different input data. The DFCN is sufficiently robust and versatile for deployment in a new setting without any requirement for re-training.

In the context of the current pandemic and the limitations of the RT-PCR test, this study introduces a model that is accurate and applicable for use in many diverse settings around the world. The AUCs achieved by our proposed DFCN on subsets A and B (0.909, 0.919) indicate the expected AUC on small subsets of clinically important parameters. These AUC values are in line with that reported in [11] (0.910) and represent a clinically relevant performance and a means to rapidly and accurately identify COVID-19 subjects.

Our study has some limitations. Since we had a limited number of training samples (382 positives, 258 negatives), it was only practical to train the last layer of a pre-trained convolutional neural network for chest x-ray interpretation. If sufficient data was available it would be preferable to train all layers. The limitation on data also limited the ways we could combine the chest x-ray model with the laboratory parameters model. Similarly, our test dataset had a relatively small number of samples (283 positives, 193 negatives). Another limitation of this study is the accuracy of the RT-PCR test which we use as our reference standard. This test has been shown to suffer from limited sensitivity in particular [34].

Future work on this topic should investigate improvements using additional data and alternative ways to combine the chest x-ray interpretation model with the diagnostic network. For full clinical validation, testing on a large diverse dataset is also essential, however in this work we have focused on novel contributions in the machine-learning field, to deal with missing data in particular.

This study has described the development and validation of a novel architecture, DFCN, robust to missing data, and validated through extensive experimentation for the task of detecting COVID-19 subjects based on laboratory and imaging parameters. This deep learning technique is ideally suited to tasks in any domain where not all input parameters are reliably available, or differ between installation settings.

## Supporting information

**S1 File.**
(PDF)

## Author Contributions

**Conceptualization:** Erdi Çallı, Keelin Murphy, Bram van Ginneken.

**Data curation:** Steef Kurstjens, Tijs Samson, Robert Herpers, Henk Smits, Matthieu Rutten.

**Funding acquisition:** Bram van Ginneken.

**Methodology:** Erdi Çallı, Keelin Murphy.

**Software:** Erdi Çallı.

**Supervision:** Keelin Murphy, Bram van Ginneken.

**Visualization:** Erdi Çallı.

**Writing – original draft:** Erdi Çallı.

**Writing – review & editing:** Keelin Murphy, Bram van Ginneken.

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
