## [Decision Letter · Decision Letter 0]

11 Jun 2021

PONE-D-21-02672

Deep Learning with robustness to missing data: A novel approach to the detection of COVID-19

PLOS ONE

Dear Dr. Çallı,

Thank you for submitting your manuscript to PLOS ONE. After careful consideration, we feel that it has merit but does not fully meet PLOS ONE’s publication criteria as it currently stands. Therefore, we invite you to submit a revised version of the manuscript that addresses the points raised during the review process.

As you will see from the reviewers' comments, there are several issues to be addressed, mainly in the experimental part. 

We look forward to receiving your revised manuscript.

Kind regards,

Ruxandra Stoean

Academic Editor

PLOS ONE

Journal Requirements:

2. Thank you for providing the date(s) when patient medical information was initially recorded. Please also include the date(s) on which your research team accessed the databases/records to obtain the retrospective data used in your study.

'Funding for this study was partially provided by the Botnar Research Centre for Child Health.'

'The author(s) received no specific funding for this work.'

5. Please ensure that you refer to Figure 7 in your text as, if accepted, production will need this reference to link the reader to the figure.

6. We note you have included a table to which you do not refer in the text of your manuscript. Please ensure that you refer to Table 4 in your text; if accepted, production will need this reference to link the reader to the Table.

7. Please include captions for your Supporting Information files at the end of your manuscript, and update any in-text citations to match accordingly. Please see our Supporting Information guidelines for more information: http://journals.plos.org/plosone/s/supporting-information

Reviewers' comments:

Reviewer's Responses to Questions

**Comments to the Author**

1. Is the manuscript technically sound, and do the data support the conclusions?

Reviewer #1: Partly

Reviewer #2: Yes

2. Has the statistical analysis been performed appropriately and rigorously? 

Reviewer #1: No

Reviewer #2: Yes

3. Have the authors made all data underlying the findings in their manuscript fully available?

Reviewer #1: No

Reviewer #2: Yes

4. Is the manuscript presented in an intelligible fashion and written in standard English?

Reviewer #1: No

Reviewer #2: Yes

5. Review Comments to the Author

Reviewer #1: I have a few observations regarding this submission before accept.

Comment #1: How does the proposed system handle missing data? A mathematical proof is needed.

Comment #2: A proof to show the robustness of the architecture is highly needed.

Comment #3: COVID-19 researches from different directions must be cited to properly present the background of the study.

Shah Muhammad Azmat Ullah, Md. Milon Islam, Saifuddin Mahmud, Sheikh Nooruddin, S. M. Taslim Uddin Raju and Md. Rezwanul Haque, “Scalable Telehealth Services to Combat Novel Coronavirus (COVID-19) Pandemic” SN Computer Science, Springer, vol. 2, no.1, pp. 18, 2020.

Md. Milon Islam, Saifuddin Mahmud, L. J. Muhammad, Md. Rabiul Islam, Sheikh Nooruddin and Safial Islam Ayon, “Wearable Technology to Assist the Patients Infected with Novel Coronavirus (COVID-19)," SN Computer Science, Springer, vol. 1, no. 6, pp. 320, Sep. 2020.

Md. Milon Islam, Shah Muhammad Azmat Ullah, Saifuddin Mahmud and S. M. Taslim Uddin Raju "Breathing Aid Devices to Support Novel Coronavirus (COVID-19) Infected Patients," SN Computer Science, Springer, vol. 1, no. 5, pp. 274, Aug. 2020.

Mohammad Marufur Rahman, Md. Motaleb Hossen Manik, Md. Milon Islam, Saifuddin Mahmud and Jong-Hoon Kim," An Automated System to Limit COVID-19 Using Facial Mask Detection in Smart City Network IEEE International IOT, Electronics and Mechatronics Conference (IEMTRONICS), IEEE, Vancouver, BC, Canada, pp. 1-5, 9-12 Sep., 2020.

Comment #4: Novelty is confusing. A highlight is required. The main contributions of the manuscript are not clear. The main contributions of the ‎article must be very clear and would be better if summarize ‎them into 3-4 points at the ‎end of the introduction.‎

Comment #5: The following references must be cited in Introduction section to describe the deep learning based systems for COVID-19 diagnosis.

M. M. Islam, F. Karray, R. Alhajj and J. Zeng, "A Review on Deep Learning Techniques for the Diagnosis of Novel Coronavirus (COVID-19)," in IEEE Access, doi: 10.1109/ACCESS.2021.3058537.

Amanullah Asraf, Md. Zabirul Islam, Md. Rezwanul Haque and Md. Milon Islam," Deep Learning Applications to Combat Novel Coronavirus (COVID-19) Pandemic," SN Computer Science, Springer, vol. 1, no. 6, pp. 363, Nov. 2020.

Muhammad Lawan Jibril, Md. Milon Islam, Usman Sani Sharif and Safial Islam Ayon, “Predictive Data Mining Models for Novel Coronavirus (COVID-19) Infected Patients Recovery,” SN Computer Science, Springer, vol. 1, no. 4, pp. 206, Jun. 2020.

Comment #6: Methodology is not clear. Provide an algorithm and flowchart of the whole work. The authors need to add a new figure to show the main structure of the proposed system. ‎This will help the reader to get a better understanding of what is going on in the proposed ‎system.‎

Comment #7: Comparison with the following works are highly required.

Md. Zabirul Islam, Md. Milon Islam and Amanullah Asraf, "A Combined Deep CNN-LSTM Network for the Detection of Novel Coronavirus (COVID-19) Using X-ray Images," Informatics in Medicine Unlocked, Elsevier, vol. 20, pp. 100412, Aug. 2020.

Prottoy Saha, Muhammad Sheikh Sadi and Md. Milon Islam," EMCNet: Automated COVID-19 Diagnosis from X-ray Images using Convolutional Neural Network and Ensemble of Machine Learning Classifiers," Informatics in Medicine Unlocked, Elsevier, vol. 22, pp. 100505, Jan. 2021.

M. M. Islam, M. Z. Islam, A. Asraf, and W. Ding, “Diagnosis of COVID-19 from X-rays Using Combined CNN-RNN Architecture with Transfer Learning,” Aug. 2020. [Online]. Available: https://www.medrxiv.org/content/10.1101/2020.08.24.20181339v1.

Comment #8: Abstract is unnecessarily wordy. Make it brief and concise. Also, Conclusion should clearly state the outcome. Some of the obtained results need to be highlighted in the conclusion section.‎

Comment #9: There are lots of typos. English needs to revise again with a professional editing service. Also, the figures are not clear in some cases.

Comment #10: Mention the limitations and future works of the developed system elaborately.

Comment #11: Please, add a small paragraph to describe the main structure of the manuscript at the end ‎of the introduction.‎

Comment #12: All the figures presented in the experimental results section are not discussed by the ‎authors. This is a critical issue in this section.‎

Comment #13: 10-fold cross-validation is adopted, right? Authors should state this in the main text explicitly.

Comment # 14: Experiments with benchmark dataset (J. P. Cohen, P. Morrison, and L. Dao, “COVID-19 Image Data Collection,” 2020, arXiv: 2003.1159. [Online]. Available: https://arxiv.org/abs/2003.11597) are highly needed to proof the universal use cases.

Reviewer #2: the Subject was interesting and I prefer to publish it but there is a minor Revision that I am not convinced about the performance of the models. the authors don't mention anything about Overfitting the model. the ROC chart doesn't show in the manuscript and it's an important problem. In table 3 (Comparison of models on the test dataset) it's better to show the models chart to compare their AUC and another metric. it's conventional to show details of confusion matrix metrics including(Sensitivity and Specificity ).

6. PLOS authors have the option to publish the peer review history of their article (what does this mean?). If published, this will include your full peer review and any attached files.

Reviewer #1: **Yes: **Md. Milon Islam

Reviewer #2: **Yes: **Mustafa Ghaderzadeh

---

## [Author Response · Author response to Decision Letter 0]

28 Jun 2021

Dear Editor and Reviewers, 

Thank you for taking the time to review our submission. In the file named "Response to reviewer comments Predicting COVID-19 DFCN.pdf" we have provided the responses to each of your comments. We are looking forward to hearing from you soon.

Kind regards,

Erdi Çallı

---

## [Decision Letter · Decision Letter 1]

14 Jul 2021

Deep Learning with robustness to missing data: A novel approach to the detection of COVID-19

PONE-D-21-02672R1

Dear Dr. Çallı,

We’re pleased to inform you that your manuscript has been judged scientifically suitable for publication and will be formally accepted for publication once it meets all outstanding technical requirements.

Kind regards,

Ruxandra Stoean

Academic Editor

PLOS ONE

Additional Editor Comments (optional):

Reviewers' comments:

Reviewer's Responses to Questions

**Comments to the Author**

1. If the authors have adequately addressed your comments raised in a previous round of review and you feel that this manuscript is now acceptable for publication, you may indicate that here to bypass the “Comments to the Author” section, enter your conflict of interest statement in the “Confidential to Editor” section, and submit your "Accept" recommendation.

Reviewer #1: (No Response)

Reviewer #2: All comments have been addressed

2. Is the manuscript technically sound, and do the data support the conclusions?

Reviewer #1: No

Reviewer #2: Yes

3. Has the statistical analysis been performed appropriately and rigorously? 

Reviewer #1: No

Reviewer #2: Yes

4. Have the authors made all data underlying the findings in their manuscript fully available?

Reviewer #1: No

Reviewer #2: Yes

5. Is the manuscript presented in an intelligible fashion and written in standard English?

Reviewer #1: No

Reviewer #2: Yes

6. Review Comments to the Author

Reviewer #1: The authors did not address all of my previous comments clearly. Even they have skipped some comments. The response is in shallow form that violate the rule for high quality journal. Actually, the revision in it's current form do not hold the criteria for publication. Another round of review of my previous comments is highly required to get the manuscript in publishable format.

Reviewer #2: this study is worthwhile and it can be published ,

the subject is interesting and applicable.

the can be better but in current form and format is acceptable and can be published in PLOSE ONE

7. PLOS authors have the option to publish the peer review history of their article (what does this mean?). If published, this will include your full peer review and any attached files.

Reviewer #1: No

Reviewer #2: **Yes: **Mustafa Ghaderzadeh

---

## [Editor Report · Acceptance letter]

21 Jul 2021

PONE-D-21-02672R1 

Deep Learning with robustness to missing data:A novel approach to the detection of COVID-19 

Dear Dr. Çallı:

I'm pleased to inform you that your manuscript has been deemed suitable for publication in PLOS ONE. Congratulations! Your manuscript is now with our production department. 

Kind regards, 

on behalf of

Dr. Ruxandra Stoean 

Academic Editor

PLOS ONE